# Adaptive Estimation of Spatial Clutter Measurement Density Using Clutter Measurement Probability for Enhanced Multi-Target Tracking

**DOI:** 10.3390/s20010114

**Published:** 2019-12-23

**Authors:** Seung Hyo Park, Sa Yong Chong, Hyung June Kim, Taek Lyul Song

**Affiliations:** Department of Electronic Systems Engineering, Hanyang University, Ansan 15588, Korea; gyeonwoo4@naver.com (S.H.P.); syong0329@hanmail.net (S.Y.C.);

**Keywords:** data association, clutter measurement density, spatial clutter measurement density estimator, multi-target tracking

## Abstract

The point detections obtained from radars or sonars in surveillance environments include clutter measurements, as well as target measurements. Target tracking with these data requires data association, which distinguishes the detections from targets and clutter. Various algorithms have been proposed for clutter measurement density estimation to achieve accurate and robust target tracking with the point detections. Among them, the spatial clutter measurement density estimator (SCMDE) computes the sparsity of clutter measurement, which is the reciprocal of the clutter measurement density. The SCMDE considers all adjacent measurements only as clutter, so the estimated clutter measurement density is biased for multi-target tracking applications, which may result in degraded target tracking performance. Through the study in this paper, a major source of tracking performance degradation with the existing SCMDE for multi-target tracking is analyzed, and the use of the clutter measurement probability is proposed as a remedy. It is also found that the expansion of the volume of the hyper-sphere for each sparsity order reduces the bias of clutter measurement density estimates. Based on the analysis, we propose a new adaptive clutter measurement density estimation method called SCMDE for multi-target tracking (MTT-SCMDE). The proposed method is applied to multi-target tracking, and the improvement of multi-target tracking performance is shown by a series of Monte Carlo simulation runs and a real radar data test. The clutter measurement density estimation performance and target tracking performance are also analyzed for various sparsity orders.

## 1. Introduction

Signals with strength higher than the detection threshold of the sensor are used as measurements for track initiation and track state update of target tracking. These measurements include not only the target measurements, but also clutter measurements due to environmental factors. Since the source of the measurements in the tracking system is not known in advance, target tracking performance may be significantly degraded if measurements generated by clutter are used when the track state is updated. It is essential to use a tracking algorithm based on data association that can statistically distinguish target and clutter measurements in a cluttered environment [1,2,3,4].

Since the number of targets existing in the surveillance region and information on the appearance and disappearance of the target cannot be known in advance, it is important to have a means for determining whether the target is being tracked by a tracking algorithm. For target tracking with track management, integrated probabilistic data association (IPDA) [5,6] and integrated track splitting (ITS) [7,8,9] have been proposed as data association algorithms for single target tracking, which include a track management method that utilizes the target existence probability of each track for controlling the track status and track number or track label. Linear multitarget-IPDA (LM-IPDA) [10], joint IPDA (JIPDA) [11], and iterative JIPDA (iJIPDA) [12] have been proposed for multiple target tracking by extending IPDA and ITS.

In the aforementioned data association algorithms, it is assumed that the number of clutter measurements is Poisson distributed with a parameter called the clutter measurement density, and the clutter measurements are assumed to be uniformly distributed in the surveillance space. The clutter measurement density is defined as the mean number of clutter measurements per unit volume of the surveillance space. The clutter measurement density is an important parameter used to calculate the data association probability and the target existence probability in the data association algorithms.

If the clutter measurement density is fixed to a design value for target tracking in heterogeneous clutter environments, the error in the clutter measurement density deteriorates not only the target state estimation performance, but also the false track discrimination (FTD) performance because prior information about the clutter measurement is unknown in actual target tracking environments. For accurate and robust target tracking in these environments, it is required to estimate the clutter measurement density adaptively. Clutter measurement density estimation methods are divided into track based estimation methods and measurement based estimation methods. In addition, they are divided into single scan estimation methods and multiple scan estimation methods depending on whether the memory is used in the calculation.

The clutter map method [13,14] is a multi-scan estimation method that uses the measurements from previous scans to calculate the clutter measurement density in the current scan. It divides the surveillance region into a finite number of cells and then estimates the clutter measurement density in each cell by statistically counting the number of existing measurements in the cell during a pre-determined multiple scan period. The clutter map can reduce the influence of bias caused by the target measurements, but estimation performance is sensitive to the parameters such as the cell size and the length of multiple scan period. It is difficult to apply the clutter map when the number of measurements and the spatial probability distribution are time varying.

In [15,16], the clutter measurement density estimation method based on the probability hypothesis density (PHD) filter [17] was handled in conjunction with a target tracking algorithm based on data association. It was designed as a feedback structure that used the intensity of clutter estimated through PHD. However, since the clutter generator is assumed to be a Gaussian function with unknown mean and unknown covariance, it is difficult to use in practical implementations due to heavy computational loads. The work in [18] proposed an interactive clutter measurement density estimator (ICMDE) based on a Gaussian mixture PHD (GM-PHD) filter [19] to estimate the clutter measurement density adaptively in environments where the clutter measurement densities are nonuniform and time varying. In [18], the Gaussian model for the clutter generator was assumed to have a known covariance for reducing the computational loads required to calculate the updated state PHD. By dividing the entire surveillance area, the clutter generator for each partition is represented as a component with the Gaussian model. These processes are performed for multiple scans to generate a reliable clutter map of the surveillance area. In [20], a method of forming a clutter map as proposed by using the histogram probabilistic multi-hypothesis tracker (H-PMHT) based on expectation maximization for image target tracking with each scene composed of millions of pixels. This method forms a clutter map through many iterations until local convergence is guaranteed.

The track based and the measurement based clutter measurement density estimation methods are classified as single scan estimation methods in which the clutter measurement density of the previous scan does not affect the clutter measurement density of the current scan. The track based clutter measurement density estimation method uses the validation gate of the track and the validated measurements existing in this gate. There exist several methods such as the conditional mean estimator based on the target perceivability [21] and the maximum likelihood estimator based on the assumption of unknown, but non-random clutter measurement density [22]. The conditional mean estimator [22] requires prior knowledge of clutter measurement density so that the maximum likelihood estimator may be used as an auxiliary estimator. For the track based clutter measurement density estimation methods, different clutter measurement densities are produced for the same measurement shared by the two tracks as the size of the validation gate of each track is different. This is a drawback of the track based clutter measurement density estimators.

The spatial clutter measurement density estimator (SCMDE) [23] is a measurement based clutter measurement density estimation method that calculates the sparsity as the reciprocal of the clutter measurement density by evaluating the volume of the hyper-sphere centered at the measurement of interest and counting the number of measurements inside the volume. The number of measurements and the hyper-sphere volume are determined by the sparsity order. Unlike the track based clutter measurement density estimation methods, it produces a unique sparsity for each measurement regardless of the validation gate size of the track involved.

It was pointed out in [23] that the existing SCMDE generates the unbiased estimates of clutter measurement density when the point of interest is the target detection for single target tracking environments. It produces biased and bigger clutter measurement density estimates than the actual ones when the point of interest is a clutter detection, which results in improved target tracking performance as the data association probabilities become smaller for the clutter detection. However, when the existing SCMDE is used for multi-target tracking environments, biased clutter measurement density estimation is expected from the nature of SCMDE that all the adjacent measurements to the point of interest are considered to be clutter detections. Through the study in this paper, a major source of biased clutter measurement density estimation of the existing SCMDE for multi-target tracking environments is analyzed, and remedies to reduce the biases are proposed. The new adaptive SCMDE for multi-target tracking (MTT-SCMDE) utilizes the clutter measurement probability to take into account only the clutter measurements for improved accuracy by reducing the biases in the clutter measurement density estimation. Through the analysis, an expansion of the volume of the hyper-sphere corresponding to each sparsity order from that of the existing SCMDE is proposed for more accurate clutter measurement density estimation.

A method that takes into account clutter-originated measurements in the clutter measurement density calculation was proposed in [24]. The performance of the SCMDE algorithm for multi-target tracking was presented in [24]. In this paper, we elaborated the theoretical development by analyzing the source of biases in the MTT-SCMDE algorithm for multi-target tracking, and refined its performance by increasing the hyper-sphere volume for the measurement of interest. The improvement was based on strict analysis presented in this paper. To verify the performance of the proposed clutter measurement density estimation method, a series of simulation runs was executed in heterogeneous clutter environments, and the results were analyzed by performance comparison to check how closely the estimated clutter measurement densities followed the true clutter measurement densities for multiple targets. In addition, the clutter measurement density estimation performance and the target tracking performance were tested for various sparsity orders and various numbers of targets involved. The proposed MTT-SCMDE was also applied to a set of real radar data for performance evaluation.

The remainder of this paper is organized as follows. The stochastic models in the target tracking algorithm are described in Section 2. Section 3 derives the LM-IPDA algorithm for multi-target tracking in a cluttered environment. The SCMDE method is briefly described in Section 4. Section 5 describes the proposed clutter measurement density estimation method in detail. The clutter measurement density estimation performance and multiple target tracking performance of the proposed method are analyzed through a series of Monte Carlo simulation runs in various tracking environments, as well as a set of real radar data in Section 6, followed by the Conclusions. Performance analysis of the existing SCMDE used in multi-target tracking environments is presented in the Appendix A.

## 2. Models

The following assumptions are applied for using multi-target tracking algorithms in a cluttered environment.

The sensor has infinite resolution; each measurement is generated only from one source; and its the origin can be either a target or clutter.Each target generates at most one measurement at each scan according to the target detection probability $P_{D}$.

Superscript τ denotes a target, or the index of a track that follows the target. Target τ’s trajectory state $x_{k}^{\tau }$ is an $n_{x}\times 1$ state vector. In this paper, the dynamics of the targets from scan *k* to scan k+1 are assumed to follow a linear dynamic model in a two-dimensional (2D) plane, such as:(1)$$\begin{array}{c} x_{k+1}^{\tau }=\Phi_{k}x_{k}^{\tau }+\Gamma_{k}w_{k}, \end{array}$$
where $x_{k}$ is a 6×1 state vector consisting of the target position, velocity, and acceleration in a 2D plane, $\Phi_{k}$ is the state propagation matrix, and $\Gamma_{k}$ is the coefficient matrix of $w_{k}$, which is a white Gaussian process noise with zero-mean and covariance matrix diag(q,q). The last term of (Equation 1) is white Gaussian with zero-mean and covariance matrix $Q_{k}=q\Gamma_{k}{\left(\Gamma_{k}\right)}^{⊤}$, and its distribution is denoted by $N(0,Q_{k})$. The state propagation matrix $\Phi_{k}$ and the coefficient matrix for the process noise $\Gamma_{k}$ follow a nearly constant velocity (NCV) model or constant turn rate (CTR) model [25].

For the NCV model, the state propagation matrix and the coefficient matrix for the process noise in (Equation 1) become:(2)$$\begin{array}{c} \Phi_{k}^{NCV}=\left[\begin{array}{ccc} I_{2} & TI_{2} & O_{2}\\ O_{2} & I_{2} & O_{2}\\ O_{2} & O_{2} & O_{2} \end{array}\right]\;\;, \end{array}$$
(3)$$\begin{array}{c} \Gamma_{k}^{NCV}=\left[\begin{array}{c} \frac{T^{2}}{2}I_{2}\\ TI_{2}\\ O_{2} \end{array}\right]\;\;, \end{array}$$
where *T* is the sampling time of a discrete time interval, $I_{2}$ is a 2×2 identity matrix, $O_{2}$ is a 2×2 null matrix, and the variance of $w_{k}$ is set to be $\sigma_{a}^{2}I_{2}$ such that $w_{k}$ of the NCV model represents the acceleration uncertainty; this implies $Q_{k}^{NCV}=\sigma_{a}^{2}\Gamma_{k}^{NCV}{\left(\Gamma_{k}^{NCV}\right)}^{⊤}$. For the NCV model, the acceleration components of $x_{k}$ are set to be zero.

For the CTR model, the state propagation matrix and the coefficient matrix for the process noise in (Equation 1) become:(4)$$\begin{array}{c} \Phi_{k}^{CTR}=\left[\begin{array}{ccc} I_{2} & \frac{\sin(\Omega_{k}T)}{\Omega_{k}}I_{2} & \frac{1-\cos(\Omega_{k}T)}{\Omega_{k}^{2}}I_{2}\\ O_{2} & \cos\left(\Omega_{k}T\right)I_{2} & \frac{\sin(\Omega_{k}T)}{\Omega_{k}}I_{2}\\ O_{2} & -\Omega_{k}\sin\left(\Omega_{k}T\right)I_{2} & \cos\left(\Omega_{k}T\right)I_{2} \end{array}\right]\;\;, \end{array}$$
(5)$$\begin{array}{c} \Gamma_{k}^{CTR}=\left[\begin{array}{c} \frac{\Omega_{k}T-\sin\left(\Omega_{k}T\right)}{\Omega_{k}^{3}}I_{2}\\ \frac{1-\cos(\Omega_{k}T)}{\Omega_{k}^{2}}I_{2}\\ \frac{\sin(\Omega_{k}T)}{\Omega_{k}}I_{2} \end{array}\right]\;\;. \end{array}$$

$w_{k}$ of the CTR model represents the uncertainty in target jerk, and its variance is $\sigma_{j}^{2}I_{2}$; this implies $Q_{k}^{CTR}=\sigma_{j}^{2}\Gamma_{k}^{CTR}{\left(\Gamma_{k}^{CTR}\right)}^{⊤}$. The turn rate $\Omega_{k}$ is adaptively estimated using the target acceleration and velocity estimates while tracking.

Target measurement model $z_{k}$ is an $n_{z}\times 1$ vector, and it is expressed as:(6)$$\begin{array}{c} z_{k}={\mathrm{Hx}}_{k}+v_{k}, \end{array}$$
where H is the measurement matrix denoted by:(7)$$\begin{array}{c} H=\left[\begin{array}{ccc} I_{2} & O_{2} & O_{2} \end{array}\right]\;\;. \end{array}$$

In (Equation 6), $v_{k}$ is a white Gaussian measurement noise of the sensor with zero-mean and covariance matrix $R_{k}$.

The sensor obtains a set of measurements $Z_{k}$ at each scan *k*. $z_{k,i}$ is the *i*th measurement of $Z_{k}$, and the measurement vector of $z_{k,i}$ can be expressed by:(8)$$\begin{array}{c} z_{k,i}={\left[\begin{array}{cc} z_{k,i}^{x} & z_{k,i}^{y} \end{array}\right]}^{⊤}\;\;, \end{array}$$
where $z_{k,i}^{x}$ and $z_{k,i}^{y}$ represent the *x* and *y* positions in the 2D Cartesian coordinate system, respectively.

## 3. LM-IPDA Algorithm for Multi-Target Tracking

In a cluttered environment, multi-target tracking algorithms with data association such as global nearest neighbor (GNN) [26,27] and joint probabilistic data association (JPDA) [28,29,30] have been widely used. However, these algorithms in general do not include an FTD procedure that can distinguish the true tracks generated by the target measurements from the false tracks generated by the clutter measurements. JIPDA and LM-IPDA are multi-target tracking algorithms with FTD functions for autonomous track management. JIPDA has optimal target tracking performance for single scan data association since it probabilistically takes into account all possible events between measurements and tracks in the cluster for each scan. However, it has heavy computational loads as the number of feasible joint events to be considered increases combinatorially depending on the number of measurements and the number of tracks. In this paper, LM-IPDA instead of JIPDA is used for multi-target tracking as the computation time increases linearly commensurate with the number of targets. In LM-IPDA, the state of track τ is represented as a hybrid state that consists of the target existence event (discrete event) and the trajectory state (continuous variable) such as:(9)$$\begin{array}{c} p\left[x_{k-1}^{\tau },\chi_{k-1}^{\tau }|Z^{k-1}\right]=P\left\{\chi_{k-1}^{\tau }|Z^{k-1}\right\}p\left(x_{k-1}^{\tau }|\chi_{k-1}^{\tau },Z^{k-1}\right), \end{array}$$
where $\chi_{k-1}^{\tau }$ represents the existence event of target τ at scan k−1, and the probability density function of the target state at scan *k* satisfies:(10)$$\begin{array}{c} p\left(x_{k-1}^{\tau }|\chi_{k-1}^{\tau },Z^{k-1}\right)=N\left(x_{k-1}^{\tau };{\hat{x}}_{k-1|k-1}^{\tau },P_{k-1|k-1}^{\tau }\right). \end{array}$$

The track recursion is composed of the following steps:prediction of track state and existence probability,selection of validated measurements,calculation of modulated clutter measurement density,update of track state and existence probability.

### 3.1. Prediction of Track State and Existence Probability

The existence event of a target in the surveillance region at scan *k* is denoted by $\chi_{k}^{\tau }$ as a random event, and ${\bar{\chi }}_{k}^{\tau }$ is the complement of $\chi_{k}^{\tau }$. The existence of a target propagates by the Markov chain one model [5,6]:(11)$$\begin{array}{c} P\left\{\chi_{k-1}^{\tau }|Z^{k}\right\}=p_{11}P\left\{\chi_{k-1}^{\tau }|Z^{k-1}\right\}, \end{array}$$
where $p_{11}$ is the transition probability of target existence.

The trajectory state of each track τ is propagated using the prediction step of the Kalman filter:(12)$$\begin{array}{c} {\hat{x}}_{k|k-1}^{\tau }=\Phi_{k}{\hat{x}}_{k-1|k-1}^{\tau } \end{array}$$
(13)$$\begin{array}{c} P_{k|k-1}^{\tau }=\Phi_{k}{\hat{P}}_{k-1|k-1}^{\tau }{\left(\Phi_{k}\right)}^{⊤}+Q_{k}. \end{array}$$

### 3.2. Selection of Validated Measurements

Since it is computationally inefficient to use all measurements in the entire surveillance region, the validation gate [2,3] is generated around the track predicted position, and the track is updated using only the measurements that exist inside the gate. If the measurement $z_{k,i}$ satisfies the following equation for track τ, $z_{k,i}$ is regarded as a validated measurement. Otherwise, it is not used for updating the states of track τ.
(14)$$\begin{array}{c} {\left(z_{k,i}-H{\hat{x}}_{k|k-1}^{\tau }\right)}^{⊤}{\left(S_{k|k-1}^{\tau }\right)}^{-1}\left(z_{k,i}-H{\hat{x}}_{k|k-1}^{\tau }\right)<\tau_{G}, \end{array}$$
with:(15)$$\begin{array}{c} S_{k|k-1}^{\tau }=HP_{k|k-1}^{\tau }H^{⊤}+R_{k}, \end{array}$$
where $\sqrt{\tau_{G}}$ is the size of the validation gate. The set of validated measurements selected by track τ and the number of validated measurements are denoted by $Z_{k}^{\tau }$ and $m_{k}^{\tau }$, respectively.

### 3.3. Calculation of Modulated Clutter Measurement Density

The calculating process of the modulated clutter measurement density is a crucial part of the LM approach, which can significantly reduce the amount of computation of JIPDA, which evaluates the probabilities of all the feasible joint events that can occur in multiple target tracking for each scan.

The modulated clutter measurement density of measurement $z_{k,i}$ can be obtained by adding the influence of other tracks to $z_{k,i}$ by utilizing the probability that measurement $z_{k,i}$ is generated from other targets to the pure clutter measurement density $\rho_{k,i}$ at the position of measurement $z_{k,i}$. Let ${\tilde{\rho }}_{k,i}^{\tau }$ denote the modulated clutter measurement density, then:(16)ρ˜k,iτ=ρk,i+∑σ∈Tkσ≠τPk,iσ1−Pk,iσpk,iσ,
where $P_{k,i}^{\sigma }$ is a measure of the influence of track σ on $z_{k,i}$, and it is represented by the prior probability that $z_{k,i}$ is generated from target σ such as:(17)$$\begin{array}{c} P_{k,i}^{\sigma }=P_{D}P_{G}P\left\{\chi_{k}^{\tau }|Z^{k-1}\right\}\frac{p_{k,i}^{\sigma }}{\rho_{k,i}}/\sum_{l=1}^{m_{k}^{\sigma }}\frac{p_{k,l}^{\sigma }}{\rho_{k,l}}, \end{array}$$
where $p_{k,i}^{\sigma }$ is a likelihood function of measurement $z_{k,i}$ with respect to track σ such that:(18)$$\begin{array}{c} p_{k,i}^{\sigma }=\frac{1}{P_{G}}N\left(z_{k,i};H{\hat{x}}_{k|k-1}^{\sigma },S_{k|k-1}^{\sigma }\right), \end{array}$$
where $P_{G}$ is the gate probability [2].

The modulated clutter measurement density of measurement ${\tilde{\rho }}_{k,i}$ is used for calculating the data association probabilities in the update step of track trajectory state, as well as the update step of the target existence probability.

### 3.4. Update of Track State and Existence Probability

Using the validated measurements selected by track τ, the posterior trajectory state of track τ and the posterior target existence probability are calculated.

Let $\beta_{k,i}^{\tau }$ denote the data association probability that is conditioned on the target existence event $\chi_{k}^{\tau }$. The data association probability $\beta_{k,i}^{\tau }$ is expressed for the event $\chi_{k,i}^{\tau }$, which indicates that the *i*th validated measurement of track τ is a target measurement, and the event $\chi_{k,0}^{\tau }$, which indicates that all the validated measurements of track τ are regarded as clutter measurements.
(19)$$\begin{array}{c} \beta_{k,i}^{\tau }=P\left\{\chi_{k,i}^{\tau }|\chi_{k}^{\tau },Z^{k}\right\}=\frac{P_{D}P_{G}}{\Lambda_{k}^{\tau }}\frac{p_{k,i}^{\tau }}{{\tilde{\rho }}_{k,i}^{\tau }}, \end{array}$$
(20)$$\begin{array}{c} \beta_{k,0}^{\tau }=P\left\{\chi_{k,0}^{\tau }|\chi_{k}^{\tau },Z^{k}\right\}=\frac{1-P_{D}P_{G}}{\Lambda_{k}^{\tau }}, \end{array}$$
where $\Lambda_{k}^{\tau }$ is the measurement likelihood ratio of track τ such that:(21)$$\begin{array}{c} \Lambda_{k}^{\tau }=1-P_{D}P_{G}+P_{D}P_{G}\sum_{i=1}^{m_{k}^{\tau }}\frac{p_{k,i}^{\tau }}{{\tilde{\rho }}_{k,i}^{\tau }}. \end{array}$$

The posterior trajectory state of track τ is calculated by the total probability theorem [31] such as:(22)$$\begin{array}{c} p\left(x_{k}^{\tau }|\chi_{k}^{\tau },Z^{k}\right)=\sum_{i=0}^{m_{k}^{\tau }}p\left(x_{k}^{\tau }|\chi_{k}^{\tau },\chi_{k,i}^{\tau },Z^{k}\right)P\left\{\chi_{k,i}^{\tau }|\chi_{k}^{\tau },Z^{k}\right\}, \end{array}$$
where $p(x_{k}^{\tau }|\chi_{k}^{\tau },\chi_{k,i}^{\tau },Z^{k})$ is a single Gaussian distribution as a posterior probability density function for the target trajectory state conditioned on the facts that target τ exists and $z_{k,i}$ is the target measurement.
(23)$$\begin{array}{c} p\left(x_{k}^{\tau }|\chi_{k}^{\tau },\chi_{k,i}^{\tau },Z^{k}\right)=N\left(x_{k}^{\tau };{\hat{x}}_{k|k,i}^{\tau },P_{k|k,i}^{\tau }\right), \end{array}$$
where the conditional mean ${\hat{x}}_{k|k,i}^{\tau }$ and covariance $P_{k|k,i}^{\tau }$ satisfy:(24)$$\begin{array}{c} {\hat{x}}_{k|k,i}^{\tau }=\left\{\begin{array}{cc} {\hat{x}}_{k|k-1}^{\tau }+K_{k|k-1}^{\tau }\left(z_{k,i}-H{\hat{x}}_{k|k-1}^{\tau }\right) & i>0\\ {\hat{x}}_{k|k-1}^{\tau } & i=0 \end{array}\right. \end{array}$$
(25)$$\begin{array}{c} P_{k|k,i}^{\tau }=\left\{\begin{array}{cc} \left(I_{6}-K_{k|k-1}^{\tau }H\right)P_{k|k-1}^{\tau } & i>0\\ P_{k|k-1}^{\tau } & i=0. \end{array}\right. \end{array}$$
where $I_{n}$ denote an n×n identity matrix, and the Kalman gain $K_{k|k-1}^{\tau }$ is expressed by:(26)$$\begin{array}{c} K_{k|k-1}^{\tau }=P_{k|k-1}^{\tau }H^{⊤}{\left(S_{k|k-1}^{\tau }\right)}^{-1}. \end{array}$$

Using the data association probabilities for the validated measurements, the updated track state estimates are obtained in the form of a Gaussian mixture such as:(27)$$\begin{array}{c} {\hat{x}}_{k|k}^{\tau }=\sum_{i=0}^{m_{k}^{\tau }}\beta_{k,i}^{\tau }{\hat{x}}_{k|k,i}^{\tau } \end{array}$$
(28)$$\begin{array}{c} P_{k|k}^{\tau }=\sum_{i=0}^{m_{k}^{\tau }}\beta_{k,i}^{\tau }\left(P_{k|k,i}^{\tau }+{\hat{x}}_{k|k,i}^{\tau }{\left({\hat{x}}_{k|k,i}^{\tau }\right)}^{⊤}\right)-{\hat{x}}_{k|k}^{\tau }{\left({\hat{x}}_{k|k}^{\tau }\right)}^{⊤}. \end{array}$$

The posterior target existence probability is used as a track score for track management including confirmation and termination. It is obtained by using the prior target existence probability and the measurement likelihood ratio such as [10]:(29)$$\begin{array}{c} P\left\{\chi_{k}^{\tau }|Z^{k}\right\}=\frac{P\left\{\chi_{k}^{\tau }|Z^{k-1}\right\}\Lambda_{k}^{\tau }}{1-\left(1-\Lambda_{k}^{\tau }\right)P\left\{\chi_{k}^{\tau }|Z^{k-1}\right\}}. \end{array}$$

## 4. The Existing Spatial Clutter Measurement Density Estimator for Single Target Tracking

The clutter measurement density is an important parameter to calculate the data association probability and the posterior target existence probability for track maintenance. In particular, when the LM approach is used in a situation where multiple targets are located in the vicinity, the merging and switching phenomena of the tracks are reduced by utilizing the modulated clutter density with moderate computational loads [10]. Therefore, it is crucial to estimate the clutter measurement density properly.

The clutter measurement density is defined as the average number of measurements that exist within a unit volume. For calculating the sparsity of $z_{k,i}$, the measurements are aligned in the ascending order of distance from $z_{k,i}$. If $Y_{k}^{i}$ denotes the set of the aligned measurements such as:(30)$$\begin{array}{c} Y_{k}^{i}=\underset{l=1}{⋃}z_{k,i}^{\left(l\right)}, \end{array}$$
where $z_{k,i}^{\left(l\right)}$ is the *l*th nearest measurement from $z_{k,i}$. Let $r_{k,i}^{\left(n\right)}$ denote the distance from $z_{k,i}$ to the *n*th nearest measurement, $z_{k,i}^{\left(n\right)}$ in $Y_{k}^{i}$. Then, $r_{k,i}^{\left(n\right)}$ becomes:(31)$$\begin{array}{c} r_{k,i}^{\left(n\right)}=∥z_{k,i}^{\left(n\right)}-z_{k,i}∥. \end{array}$$

The SCMDE estimates the sparsity of the measurements, which is the reciprocal of the clutter measurement density. The sparsity of $z_{k,i}$ is obtained from:(32)$$\begin{array}{c} {\hat{\gamma }}_{k,i}^{\left(n\right)}=\frac{1}{\rho_{k,i}^{\left(n\right)}}=\frac{V\left(r_{k,i}^{\left(n\right)}\right)}{n}, \end{array}$$
where *n* and $V\left(r_{k,i}^{\left(n\right)}\right)$ denote the sparsity order and the volume of the hyper-sphere with radius $r_{k,i}^{\left(n\right)}$ for $z_{k,i}$, respectively. $V\left(r_{k,i}^{\left(n\right)}\right)$ is expressed by:(33)$$\begin{array}{c} V\left(r_{k,i}^{\left(n\right)}\right)=C_{n_{z}}{\left(r_{k,i}^{\left(n\right)}\right)}^{n_{z}}, \end{array}$$
where $n_{z}$ represents the dimension of the measurement space and $C_{n_{1}}=2$, $C_{n_{2}}=\pi$, and $C_{n_{3}}=\frac{4\pi }{3}$. Figure 1 schematically illustrates the hyper-spheres with various sparsity orders for $z_{k,i}$ in a 2D measurement space.

In the process of deriving the sparsity, track information is not used. A measurement shared by two or more tracks has a unique clutter measurement density regardless of the track states.

## 5. SCMDE for Multi-Target Tracking Environments

### 5.1. Drawbacks of the Existing SCMDE for Multi-Target Tracking

It was introduced in [23] that the SCMDE for single target tracking environments yields an accurate clutter measurement density when the point of interest is the target detection. When the point of interest is a clutter detection, the SCMDE generates a biased and smaller sparsity than the actual value, which implies a bigger clutter measurement density. This phenomenon gives benefits to single target tracking as the bigger clutter measurement density decreases the data association probability for the clutter detection in the probabilistic data association (PDA) algorithm. It was also introduced in [23] that these benefits are reduced as the sparsity order increases. Therefore, the existing SCMDE improves target tracking performance for single target tracking.

In this subsection, the performance of the existing SCMDE is analyzed for multi-target tracking in homogeneous clutter environments. The detailed derivations are given in Appendix A of this paper. When the point of interest is a target detection for a two-target case, the average value of the sparsity estimate for the point of interest $z_{k,i}$ becomes:(34)$$\begin{array}{c} E\left\{{\hat{\gamma }}_{k,i}^{\left(n\right)}\right\}=\left\{\begin{array}{cc} \frac{1}{\rho }\left(1-\frac{1-e^{-\rho V^{\left(1\right)}}}{\rho V^{\left(1\right)}}\right), & n=1\\ \frac{1}{\rho }\left(1-\frac{1-e^{-\rho V^{\left(2\right)}}}{\rho V^{\left(2\right)}}+\frac{e^{-\rho V^{\left(2\right)}}}{2}\right), & n=2 \end{array}\right. \end{array}$$
where *n* is the sparsity order, ρ is the clutter measurement density of the homogeneous clutter environment, and $V^{\left(n\right)}$ is the volume of the hyper-sphere used for the sparsity estimation. If (Equation 34) is compared to the true sparsity, $\frac{1}{\rho }$, which can be obtained from the existing SCMDE for single target tracking as shown in (35) of [23], the sparsity estimates are smaller than the true ones and biased. The bias becomes reduced as *n* increases and $V^{\left(n\right)}$ becomes bigger. In contrast to single target tracking environments, the SCMDE generates bigger clutter measurement density estimates when the point of interest is a target detection, which results in a reduced data association probability for the target detection and deteriorated target tracking performance for multi-target tracking environments.

When the point of interest is a clutter detection for two-target cases, the average value of the sparsity estimates for the point of interest, $z_{k,i}$, becomes:(35)$$\begin{array}{c} E\left\{{\hat{\gamma }}_{k,i}^{\left(n\right)}\right\}=\left\{\begin{array}{cc} \frac{1}{\rho }-\frac{1}{\rho^{2}V^{\left(1\right)}}\left(1-e^{-\rho V^{\left(1\right)}}\right), & n=1\\ \frac{1}{\rho }-\frac{1}{2\rho^{2}V^{\left(2\right)}}\left(1-e^{-\rho V^{\left(2\right)}}\right), & n=2 \end{array}\right. \end{array}$$

The average sparsity estimates in (Equation 35) are smaller than the actual $\frac{1}{\rho }$, and this fact results in bigger clutter measurement density estimates. The average sparsity estimate for n=1 in (Equation 35) is the same as n=1 for single target tracking environments specified in (23) of [23]. When n=2, the average sparsity estimate becomes bigger than n=1, and this indicates that the clutter measurement density estimates become less biased for n=2. This indicates that more accurate clutter measurement density estimation is possible with the PDA algorithm as *n* increases. From the above analysis, the existing SCMDE has two incompatible aspects in tracking performance for multi-target tracking environments. One aspect is that tracking performance becomes deteriorated as it generates smaller data association probabilities than the actual ones for true target detections. Another aspect is that tracking performance is improved as it generates smaller data association probabilities than the actual one for clutter measurements. These incompatible aspects are due to the biased and reduced sparsity estimates described in (Equation 34) and (Equation 35).

In order to improve tracking performance for multi-target tracking environments, it is more important to have the improved data association results with less biased clutter measurement density estimates. This can be done by evaluating the clutter measurement probability of each validated measurement for counting only the number of clutter measurements (excluding the number of target measurements), inside the volume of the hyper-sphere $V(r_{k,i}^{\left(n\right)})$ specified in (Equation 33). The clutter measurement probability is the probability that the measurement is a clutter detection not from a target. If the clutter measurement probability is used for the sparsity estimates, enhanced tracking is expected due to less biased clutter measurement density estimates. This has a more significant effect in performance improvement when the point of interest is a target detection rather than a clutter detection. From the analysis in this section, the magnitude of bias of the sparsity estimate of (Equation 34) and (Equation 35) becomes smaller as *n* increases and the volume of the hyper-sphere $V^{\left(n\right)}$ increases. In the next subsection, the adaptive SCMDE algorithm for multi-target tracking (MTT-SCMDE) is proposed to take into account the clutter measurement probability and increased hyper-sphere volume for each sparsity order *n* to achieve enhanced tracking performances.

### 5.2. MTT-SCMDE

To estimate the clutter measurement density accurately for these multi-target tracking environments, we propose a method to calculate the probability that adjacent measurements are generated from clutter and use this probability to estimate the clutter measurement density.

To derive the clutter measurement probability, two events are defined.

$\chi_{k,j}^{0}$: an event that the measurement $z_{k,j}$ is not a target measurement for any of the tracks at scan *k*.$\chi_{k,j}^{\tau }$: an event that the measurement $z_{k,j}$ is a target measurement originated from the target τ at scan *k*

Let $T_{k}$ denote the set of the cluster targets at scan *k*, and $H_{k}$ represent the event that $T_{k}$ exists at scan *k* such as:(36)$$\begin{array}{c} H_{k}=\underset{\sigma \in T_{k}}{⋃}\chi_{k}^{\sigma }. \end{array}$$

The probability that $z_{k,j}$ is generated from clutter under $H_{k}$, the mutual exclusiveness of $z_{k,j}$ sources, becomes:(37)$$\begin{array}{c} P\left(\chi_{k,j}^{0}|Z^{k-1},H_{k}\right)=\alpha_{k}\prod_{\sigma \in T_{k}}^{}\left(1-P_{k,j}^{\sigma }\right), \end{array}$$
where $P_{k,j}^{\sigma }$ is the prior probability that $z_{k,j}$ is generated from target σ introduced in (Equation 17) and where $\alpha_{k}$ is a normalization constant.

The probability that $z_{k,j}$ is generated from target $\tau \in T_{k}$ becomes: (38)P(χk,jτ|Zk−1,Hk)=αkPk,jτ∏η∈Tkη≠τ1−Pk,jη(39)$$\begin{array}{cc} = & \alpha_{k}\frac{P_{k,j}^{\tau }}{1-P_{k,j}^{\tau }}\prod_{\eta \in T_{k}}^{}\left(1-P_{k,j}^{\eta }\right). \end{array}$$

From (37) and (39), $\alpha_{k}$ can be obtained from the mutual exclusiveness of $z_{k,j}$ sources such as:(40)$$\begin{array}{c} P\left(\chi_{k,j}^{0}|Z^{k-1},H_{k}\right)+\sum_{\tau \in T_{k}}P\left(\chi_{k,j}^{\tau }|Z^{k-1},H_{k}\right)=1. \end{array}$$

Then, $\alpha_{k}$ is obtained as:(41)$$\begin{array}{c} \alpha_{k}=\frac{1}{\prod_{\sigma \in T_{k}}^{}\left(1-P_{k,j}^{\sigma }\right)\left(1+\sum_{\eta \in T_{k}}^{}\frac{P_{k,j}^{\eta }}{1-P_{k,j}^{\eta }}\right)}. \end{array}$$

Therefore, the clutter measurement probability $P(\chi_{k,j}^{0}|Z^{k-1},H_{k})$ in (Equation 37) can be expressed as:(42)$$\begin{array}{c} P\left(\chi_{k,j}^{0}|Z^{k-1},H_{k}\right)=\frac{1}{1+\sum_{\eta \in T_{k}}^{}\left(\frac{P_{k,j}^{\eta }}{1-P_{k,j}^{\eta }}\right)}. \end{array}$$

The proposed MTT-SCMDE utilizes the clutter measurement probabilities of the element of $Y_{k}^{i}$ defined in (Equation 30). Let $C_{k,i}^{\left(l\right)}$ be the clutter measurement probability of $z_{k,i}^{\left(l\right)}$, the *l*th nearest measurement from $z_{k,i}$. If the cumulative sum $C_{k,i}^{\left(l\right)}$ from l=1 is bigger than the predetermined sparsity order *n*, which is the expected number of clutter measurements, the summation is stopped for the sparsity calculation. If:(43)$$\begin{array}{c} \sum_{l=1}^{m-1}C_{k,i}^{\left(l\right)}<n\leq \sum_{l=1}^{m}C_{k,i}^{\left(l\right)}, \end{array}$$
then the radius $r_{k,i}^{\left(n\right)}$ for the hyper-sphere volume calculation becomes:(44)$$\begin{array}{c} r_{k,i}^{\left(n\right)}=∥z_{k,i}^{\left(m+1\right)}-z_{k,i}∥, \end{array}$$
where $z_{k,i}^{\left(m+1\right)}\in Y_{k}^{i}$.

In this paper, $z_{k,i}^{\left(n+1\right)}$ is used to calculate $r_{k,i}^{\left(n\right)}$ instead of $z_{k,i}^{\left(n\right)}$ used in [23] for single target tracking to produce less biased estimates of the clutter measurement density for multi-target tracking environments.

The estimated sparsity of order *n* becomes:(45)$$\begin{array}{c} {\hat{\gamma }}_{k,i}^{\left(n\right)}=\frac{V\left(r_{k,i}^{\left(n\right)}\right)}{\sum_{l=1}^{m}C_{k,i}^{\left(l\right)}}, \end{array}$$
where $V\left(r_{k,i}^{\left(n\right)}\right)$ is the volume of the hyper-sphere with radius $r_{k,i}^{\left(n\right)}$ defined in (Equation 44).

When estimating the sparsity, the existing SCMDE utilizes the number of measurements in the hyper-sphere, while the proposed MTT-clutter measurement density estimation method utilizes the mean number of clutter measurements with the clutter measurement probability to reduce biases in the clutter measurement density estimates in multi-target tracking applications.

Figure 2 shows an expansion of the volume of the hyper-sphere of the MTT-SCMDE if it is used for data association in single target environments. Compared to Figure 1, the volume of the hyper-sphere for each sparsity order is increased.

## 6. Performance Tests

### 6.1. Simulation Experiments

The performance test was done to compare the results with respect to FTD performance and the accuracy of estimated clutter measurement density for the cases, which utilized:True clutter measurement density (true CMD),SCMDE with various sparsity orders,MTT-SCMDE with various sparsity orders.

The sets of simulation experiments are presented for multi-target tracking in a heterogeneous environment with varying the number of targets.

The sampling time of sensor *T* was 1 s, and the measurement noise covariance was $R_{k}=25I_{2}m^{2}$. The target detection probability $P_{D}$ and the gate probability $P_{G}$ were 0.8 and 0.99, respectively. One simulation run consisted of 50 scans, and the total number of Monte Carlo simulation runs was 500. To initialize the track, the two point differencing method [14] was employed if the calculated velocity obtained from two consecutive scans was smaller than the predetermined maximum velocity constraint $V_{max}=25$ m/s.

In these simulations, the following four evaluation indices were calculated:confirmed true track rate (CTTR) [1],position root mean squared error (RMSE),track retention test statistics [10],clutter measurement density estimation performance.

The confirmed track was defined as one whose posterior target existence probability calculated by (Equation 29) was bigger than a predetermined confirmed threshold. Among confirmed tracks, the tracks satisfying the following equation were classified as the confirmed true tracks for tracking performance evaluation purposes:(46)$$\begin{array}{c} {\left(x_{k}-{\hat{x}}_{k|k}^{\tau }\right)}^{⊤}{\left(P_{0|0}^{\tau }\right)}^{-1}\left(x_{k}-{\hat{x}}_{k|k}^{\tau }\right)<\gamma_{true}, \end{array}$$
where $x_{k}$ and $P_{0|0}^{\tau }$ are the state vector of the true target and the initial error covariance matrix of the confirmed track τ, respectively. Conversely, each confirmed track met the following test, and it became a confirmed false track.
(47)$$\begin{array}{c} {\left(x_{k}-{\hat{x}}_{k|k}^{\tau }\right)}^{⊤}{\left(P_{0|0}^{\tau }\right)}^{-1}\left(x_{k}-{\hat{x}}_{k|k}^{\tau }\right)>\gamma_{false}, \end{array}$$

The confirmed true track rate (CTTR) is an evaluation index showing the statistical ratio of confirmed true tracks over time. The position root mean squared error is the distance error between the confirmed true track and the true target, and it was obtained only for the confirmed true tracks over time. The track retention test statistic is an evaluation index of the multi-target tracking algorithms and accumulates statistics on how much the confirmed true tracks are retained or lost between retention test start time (RST) and retention test end time (RET). In these simulations, RST and RET were designated to be 15 and 35, respectively. The retention test was to check the following items, and they were used to indicate the statistical ratio representing the robustness of each algorithm for the period in which the targets were located in the immediate vicinity:nCase: the total number of CTTs at RST,nOk: the percentage of nCase CTTs that still followed the original target at RET,nSwitch: the percentage of nCase CTTs that did not follow the original target at RET,nMerge: the percentage of merging two or more nCase CTTs during the retention test,nLost: the percentage of nCase CTTs that were terminated during the retention test.

The clutter measurement density estimation performance is an evaluation index of how closely the estimated clutter measurement density follows the true clutter measurement density using the clutter measurement density estimation method. The performance of the proposed algorithm was tested in comparison with the existing SCMDE algorithm for multi-target tracking by varying the number of targets in a heterogeneous clutter environment. Simulations were performed for 3, 5, and 7 targets in three scenarios, and the performance of clutter measurement density estimation was analyzed as the number of targets increased. In addition, the effectiveness and the robustness of the tracking performance were verified through a test with real radar data.

#### 6.1.1. The Number of Targets: 3

The simulation considered the 2D surveillance region depicted in Figure 3. The targets maneuvered slightly to form curved trajectories within the surveillance region. To track maneuvering targets, the LM-IPDA-interacting multiple model (LM-IPDA-IMM) [32] was employed. The LM-IPDA-IMM algorithm used in this study utilized the NCV model and the CTR model introduced in Section 2. The targets were located apart at the beginning of the scenario, then they were located in the immediate vicinity at Scan 25, and then moved away from each other.

The base clutter measurement density was $1\times 10^{-4}$ scans/m${}^{2}$, and it increased to $3\;\times \;10^{-4}$ scans/m${}^{2}$ in the high clutter measurement density region; clutter measurements were spatially distributed with a uniform distribution inside each cluttered region for every scan. In Figure 3, the squares represent the measurements of each target. The gray symbols represent the clutter measurements generated during a single simulation run.

Figure 4 represents the CTTR for three targets in 500 Monte Carlo runs, and the position RMSE for Target 1 and the estimated clutter measurement density for Target 1 over time are listed in Figure 5 and Figure 6, respectively. For fair comparisons, the number of confirmed false tracks of each case was made to be almost 40 for all 500 Monte Carlo simulation runs by adjusting the initial target existence probability while the confirmation threshold was equal for all the algorithms in comparison. Using the true clutter measurement density showed that the CTTR had the fastest build-up. Even if the same sparsity order was applied, the proposed clutter measurement density estimation method provided better tracking results than the SCMDE. The closer to the true clutter measurement density the estimated clutter measurement density was, the better the performance was. At around Scan 25, when the targets were located in the immediate vicinity, the SCMDE estimated the clutter measurement density of the target measurement, which appeared to be bigger than the actual. This resulted in a slow build-up of the CTTR. By comparing the CTTR results for the sparsity order of n=1 and n=5 for the same clutter measurement density estimation methods, one could find that higher sparsity order resulted in better tracking performance because the higher the sparsity order was, the more accurate the estimated clutter measurement density was, as shown in Figure 6. The position RMSEs shown in Figure 5 were calculated for only the confirmed true tracks, which satisfied (Equation 46) such that the RMSEs looked similar in the order of magnitudes for all the algorithms in comparison as the confirmed true tracks passed the condition of (Equation 46). However, the number of samples involved in the RMSE calculation was quite different for each algorithms, as shown by the CTTR of Figure 4, which implied high reliability in RMSE for the algorithms with high CTTR and low reliability in RMSE for the algorithms with low CTTR. Figure 7 shows the true states and the estimated states of Target 1 over time for the position, velocity, and acceleration elements of each coordinate axis. Only the averaged state estimates of the confirmed tracks are shown in Figure 7. The existing SCMDE with the sparsity order of n=1 showed the worst estimation performance among the algorithms in comparison. The target tracking algorithm using the proposed MTT-SCMDE with the sparsity order of n=5 showed similar estimation performance to the one using the true clutter measurement density, and its state estimates were close to the true target states. This implied that the proposed MTT-SCMDE produced more reliable and accurate estimates for multi-target tracking than the existing SCMDE.

Although the clutter measurement density was estimated close to the actual for the proposed method with sparsity order of n=5, the tracking performance was slightly worse than using the true clutter measurement density. It produced the best tracking performance among the methods in comparison. Therefore, the proposed method with a high sparsity order was a viable solution for this environment.

Table 1 shows the statistics of the track retention test. The proposed clutter measurement density estimation method had a higher track maintenance performance in terms of true track confirmation and track losses including switch and merge than the SCMDE method with the same sparsity order.

#### 6.1.2. The Number of Targets: 5

In this scenario, we analyzed the clutter measurement density estimation performance by increasing the number of targets to five, as shown in Figure 8. The parameters except the number of targets were the same as in the previous scenario. The number of confirmed false tracks was made almost equal as in the previous scenario by adjusting the initial target existence probability.

Figure 9, Figure 10 and Figure 11 represent CTTR, position RMSE for Target 1, and the estimated clutter measurement density for Target 1 over time for the scenario, respectively. All the algorithms had the same trend in estimation performance as in the previous scenario. The proposed clutter measurement density estimation method with the sparsity order of n=5 showed the best tracking performance among the methods in comparison because it estimated the clutter measurement density similar to the true clutter measurement density even if the number of closely located targets increased. As shown in Table 2, nCase and nOk for the MTT-SCMDE with n=5 represented the best tracking performance among the adaptive estimation methods in comparison.

#### 6.1.3. The Number of Targets: 7

The measurement histories of the seven closely located targets are shown in Figure 12. In this scenario, considering that the number of targets was seven, the simulation was performed by extending the sparsity order to 7 in addition to the 1 and 5 used in the previous scenarios. As in the previous scenarios, multiple targets were gathered in the high density clutter region.

Figure 13 shows the CTTR over time. As shown in Figure 14, the estimation errors of with the sparsity order n=7 were similar to the result using the true clutter measurement density. As the number of targets increased, increasing the sparsity order implied that better tracking results could be obtained, and the proposed MTT-SCMDE had better tracking performance compared to the existing SCMDE with the same sparsity order. Figure 15 represents the estimated clutter measurement density over time and shows that even with a large number of closely located targets, the proposed method had the best performance of estimating the clutter measurement density. In Table 3, the MTT-SCMDE with n=7 showed more than 80% track retention performance, similar to the case with true clutter measurement density. It showed the best tracking performance among the adaptive estimation algorithms in comparison.

### 6.2. Test with Real Radar Data

In this section, a set of measurements obtained from a surveillance radar system is utilized for performance analysis of the proposed algorithm. The main focus of the analysis was to verify the robustness of the algorithm for tracking in clutter without track loss and switching, especially in the region where the multiple targets were located in the vicinity. For this data gathering experiment, there were no other reference sensors to measure the exact locations of the target. Therefore, it was not possible to analyze the accuracy of the target tracking, so we focused on the maintenance performance for the confirmed tracks and the discrimination performance for the false tracks caused by clutter.

The radar acquired measurements every one second. The 2D radar measurements consisted of distance and azimuth information. The measurements of the distance and azimuth information were converted to the x,y positions in the Cartesian coordinate system for the tracking algorithms. LM-IPDA with the NCV model in Section 2 was used for tracking in this performance test, and the results of target tracking were compared for three cases, which employed a fixed value $(1\;\times \;10^{-7}$ scan/m${}^{2})$ for the clutter measurement density, adaptive clutter measurement density estimation with the existing SCMDE, and the proposed MTT-SCMDE.

The initial target existence probability of the track was set to be 0.1. When the target existence probability of track was smaller than $\frac{1}{10}$ of the initial value, the track would be terminated, and if the target existence probability was bigger than 0.95, it was classified as a confirmed track.

Figure 16 contains the measurement dataset for the entire period of 92 s. As shown in Figure 16, the radar detection range was 90 km, and the radar measurements were used from −90${}^{\circ }$ to 0${}^{\circ }$ from the north. The gray symbols represent the measurements obtained from the radar.

Figure 17, Figure 18 and Figure 19 show the trajectories of the confirmed tracks estimated by the LM-IPDA algorithm with the NCV model, which utilized fixed clutter measurement density, the SCMDE, and the MTT-SCMDE, respectively. The sparsity order n=5 was used for the SCMDE and the MTT-SCMDE. The main difference in the tracking results of the three cases was shown for the two targets in a formation flight in the high clutter measurement density region, which was specified by a green circle of each figure. In the case of using the fixed clutter measurement density, no confirmed track was generated for the left of the two targets in a formation flight. When the SCDME was used, the tracks for both targets were confirmed in the beginning, but one of the confirmed tracks was lost as the distance between the two targets became smaller. As the SCMDE did not distinguish the nature of adjacent measurements when estimating the clutter measurement density, a bias in the clutter measurement density estimates was included for the closely located targets, and this bias decreased the data association probability of the true target measurement. This resulted in the loss of the confirmed track. However, in the case of the proposed MTT-SCMDE, it can be seen from Figure 19 that the tracks for both targets were confirmed without loss of tracks. This demonstrated the robustness of the proposed MTT-SCMDE algorithm in practical applications.

## 7. Conclusions

The clutter measurement density is a parameter required to calculate the data association probability of the measurement and target existence probability of a track and has a large impact on target tracking performance even with small changes. This paper presented the SCMDE with clutter measurement probability to estimate the clutter measurement density adaptively for non-parametric multi-target tracking in environments where there is no prior information about clutter distribution. The algorithm was developed by analyzing the causes of estimation performance deterioration of the existing SCMDE. The proposed clutter measurement density estimation method calculated the sparsity of the measurements by probabilistically classifying adjacent measurements as a target measurement or as a clutter measurement. We demonstrated the effectiveness of the proposed clutter measurement density estimation method, which was designed to achieve more accurate and robust clutter measurement density estimation by showing the performance improvement for multi-target tracking through simulation studies in various environments and a test with real radar data.

## Figures and Tables

**Figure 1 sensors-20-00114-f001:**
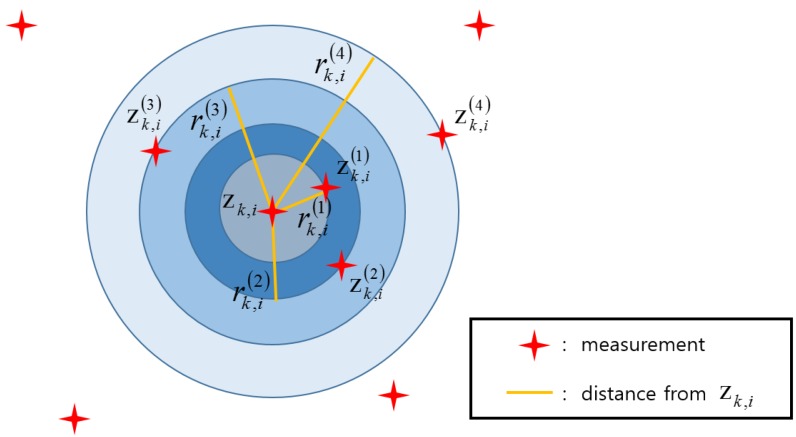
Hyper-spheres for the existing spatial clutter measurement density estimator (SCMDE) in a 2D space.

**Figure 2 sensors-20-00114-f002:**
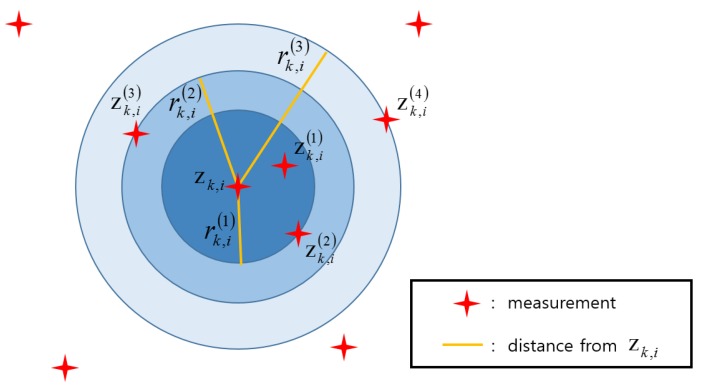
Hyper-spheres for the multi-target tracking (MTT)-SCMDE used for single target tracking in a 2D space.

**Figure 3 sensors-20-00114-f003:**
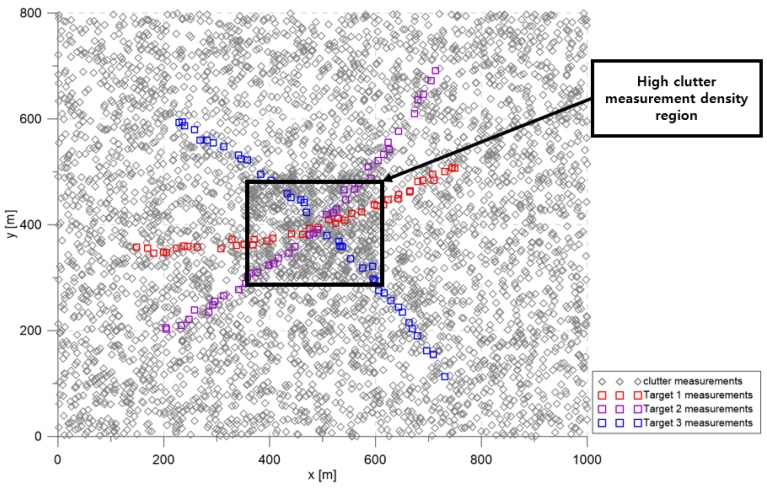
Simulation scenario with three targets.

**Figure 4 sensors-20-00114-f004:**
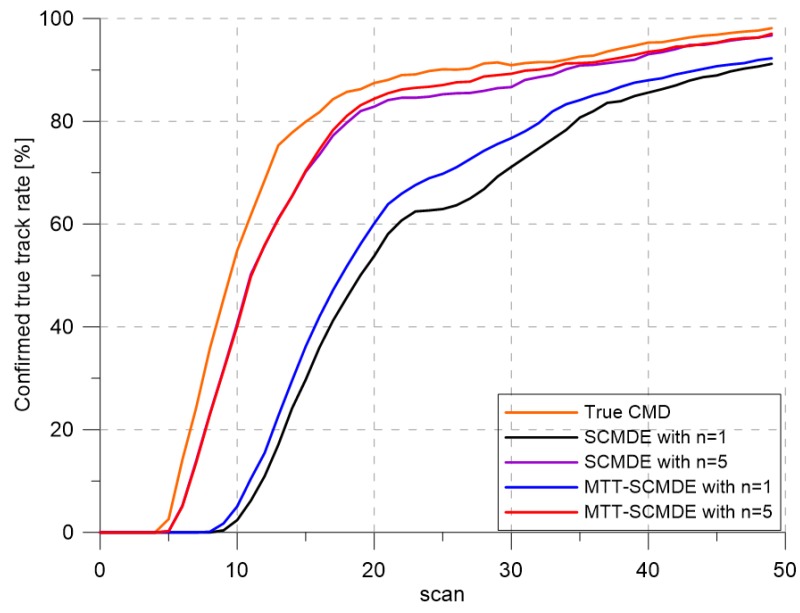
Confirmed true track rate. CMD, clutter measurement density.

**Figure 5 sensors-20-00114-f005:**
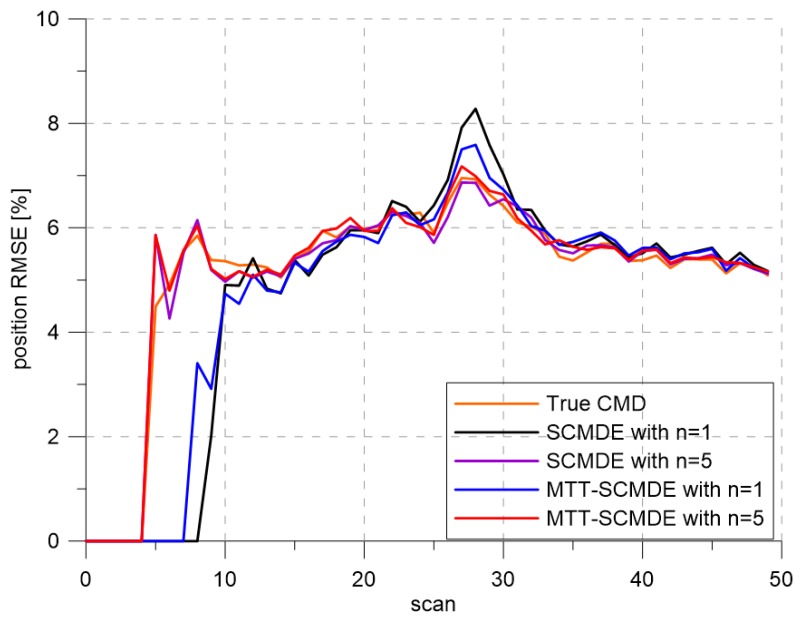
Position RMSE.

**Figure 6 sensors-20-00114-f006:**
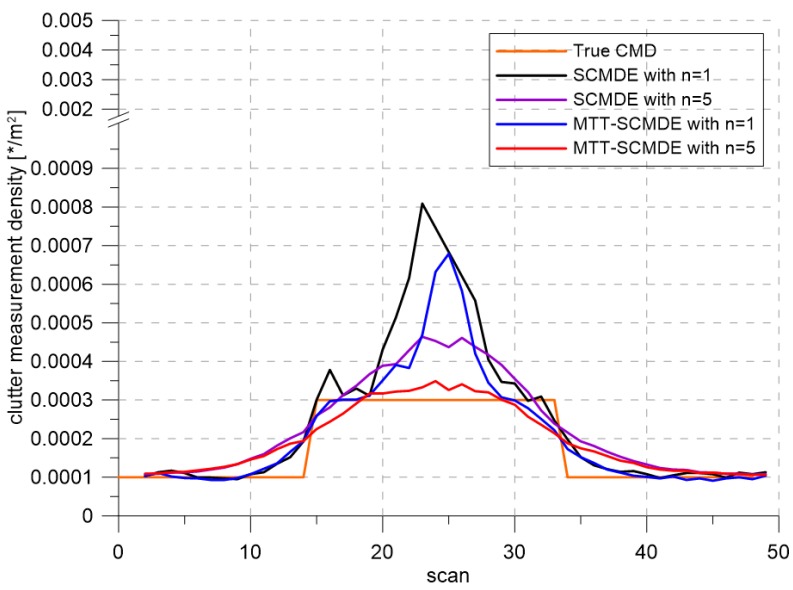
True clutter measurement density and estimated clutter measurement density.

**Figure 7 sensors-20-00114-f007:**
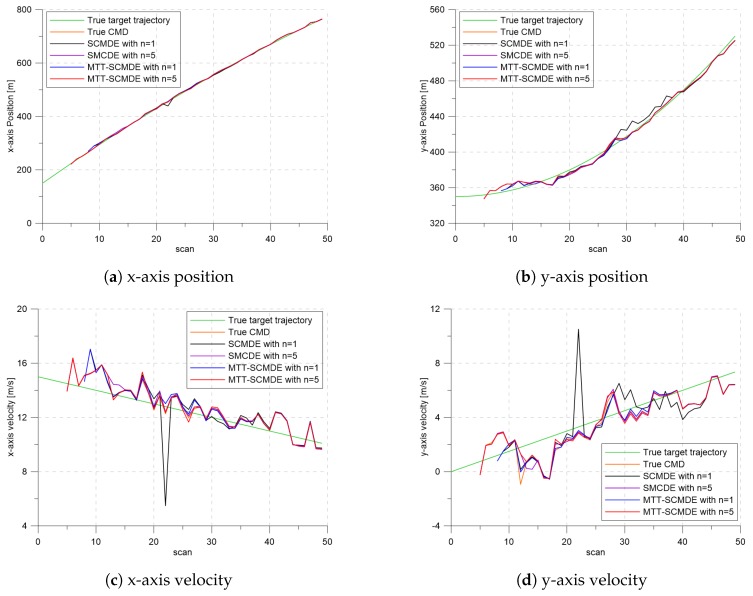
The true states and the estimated states of Target 1 over time.

**Figure 8 sensors-20-00114-f008:**
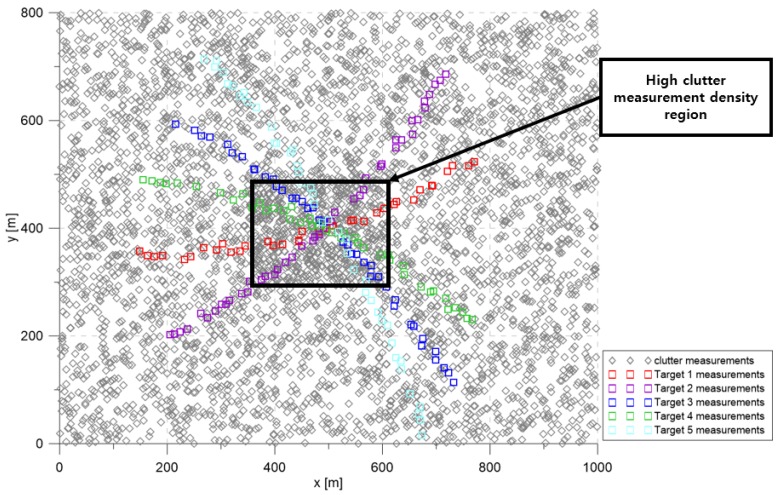
Simulation scenario with five targets.

**Figure 9 sensors-20-00114-f009:**
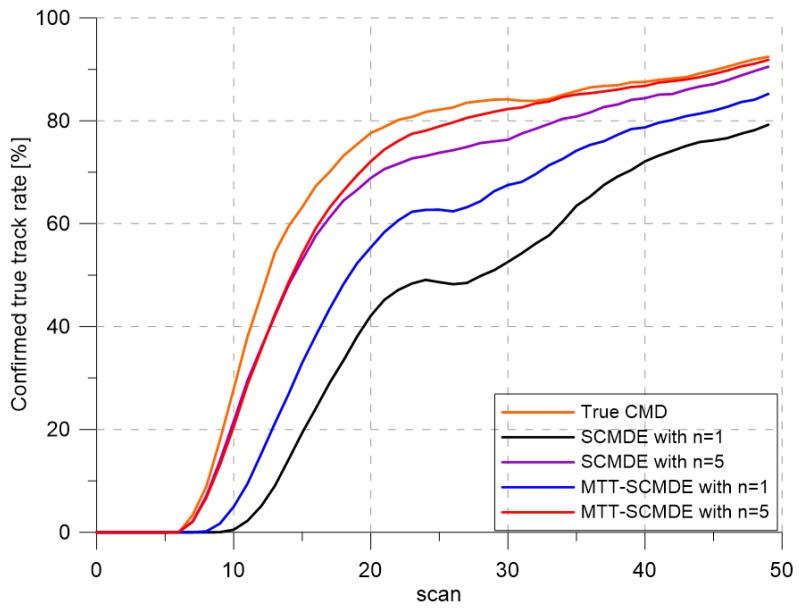
Confirmed true track rate.

**Figure 10 sensors-20-00114-f010:**
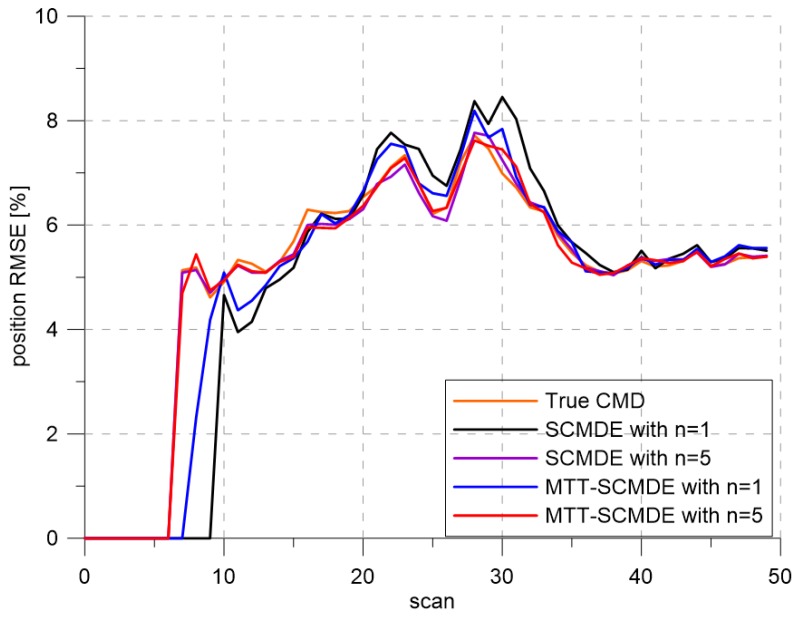
Position RMSE.

**Figure 11 sensors-20-00114-f011:**
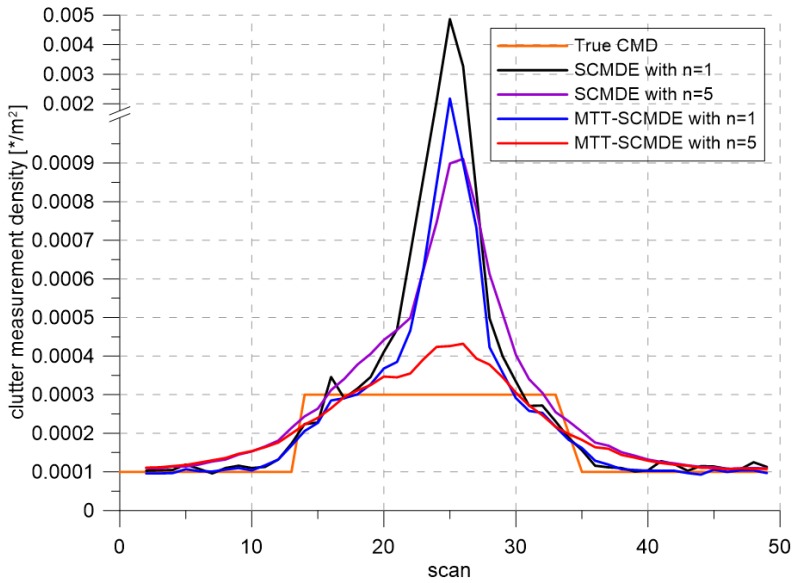
True clutter measurement density and estimated clutter measurement density.

**Figure 12 sensors-20-00114-f012:**
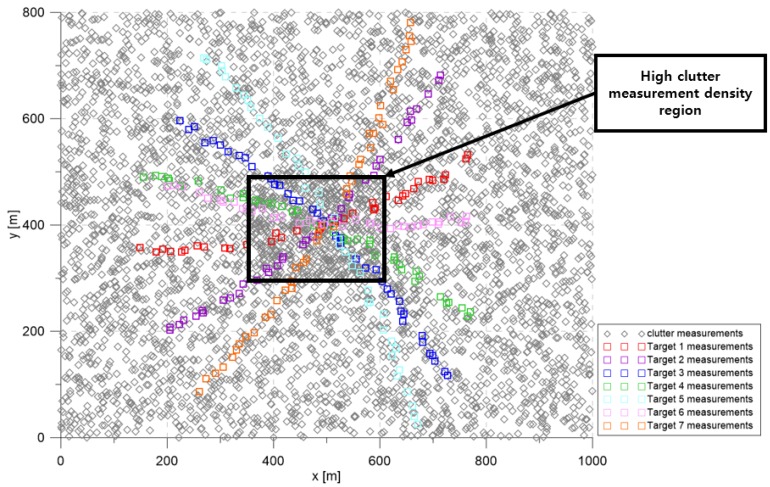
Simulation scenario with seven targets.

**Figure 13 sensors-20-00114-f013:**
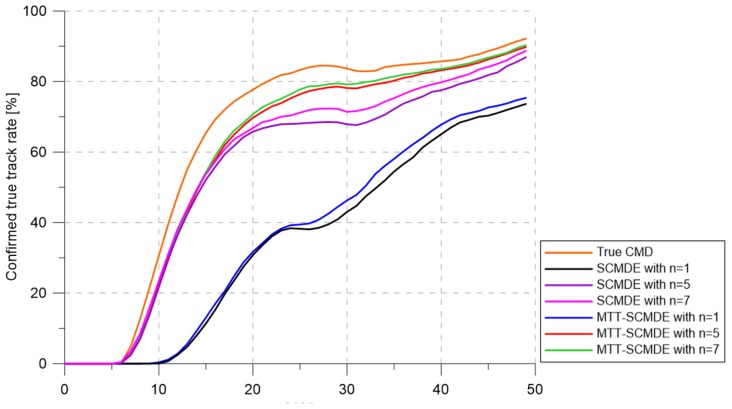
Confirmed true track rate.

**Figure 14 sensors-20-00114-f014:**
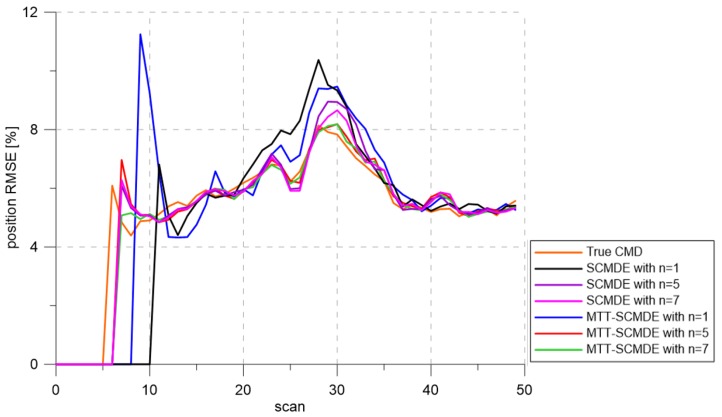
Position RMSE.

**Figure 15 sensors-20-00114-f015:**
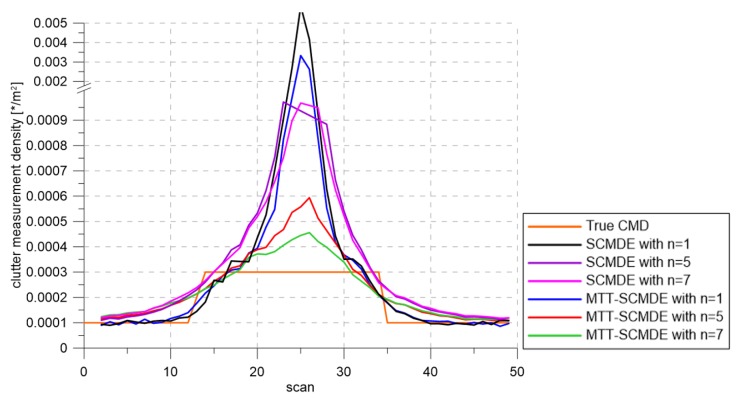
True clutter measurement density and estimated clutter measurement density.

**Figure 16 sensors-20-00114-f016:**
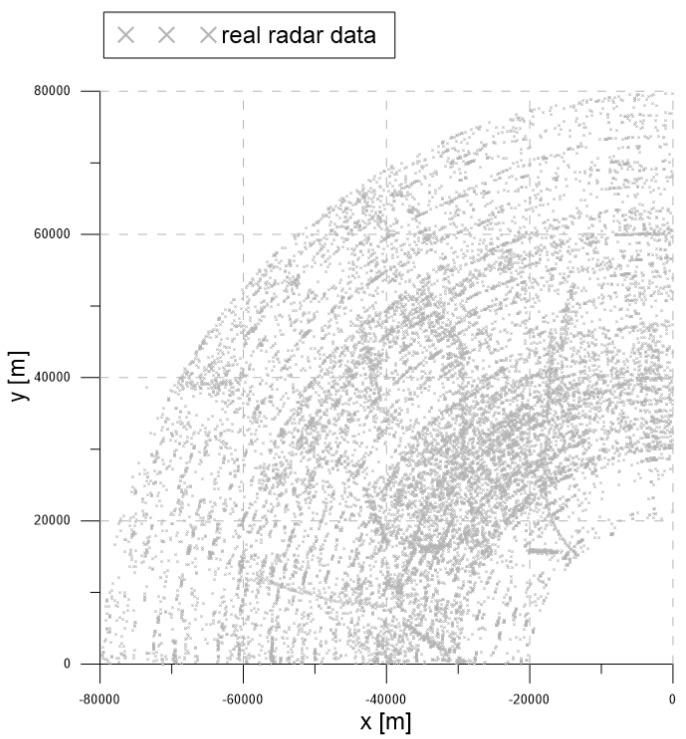
Real radar measurements within the surveillance region.

**Figure 17 sensors-20-00114-f017:**
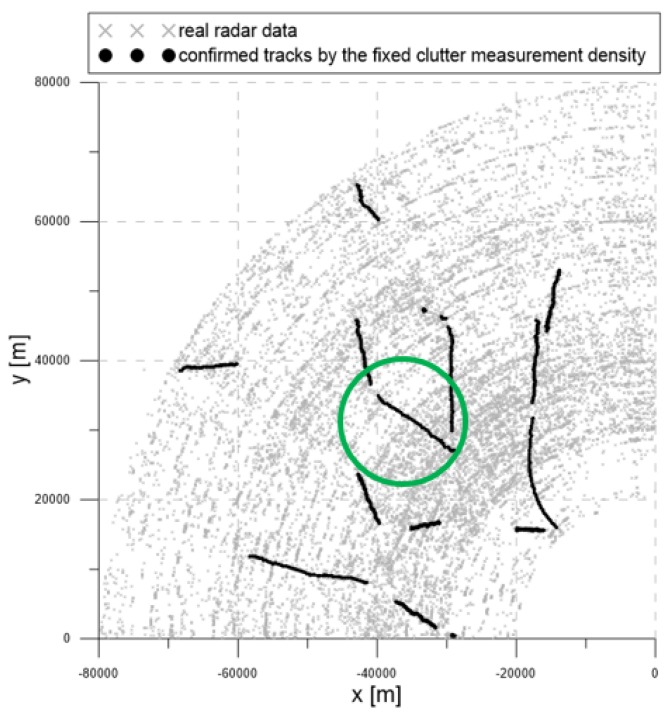
The trajectories of the confirmed tracks by using the fixed clutter measurement density.

**Figure 18 sensors-20-00114-f018:**
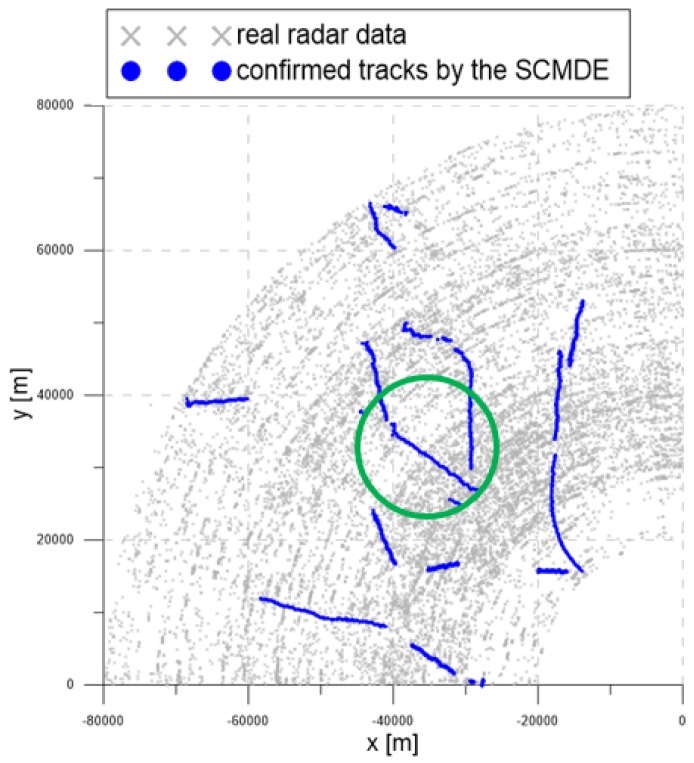
The trajectories of the confirmed tracks by using the SCMDE.

**Figure 19 sensors-20-00114-f019:**
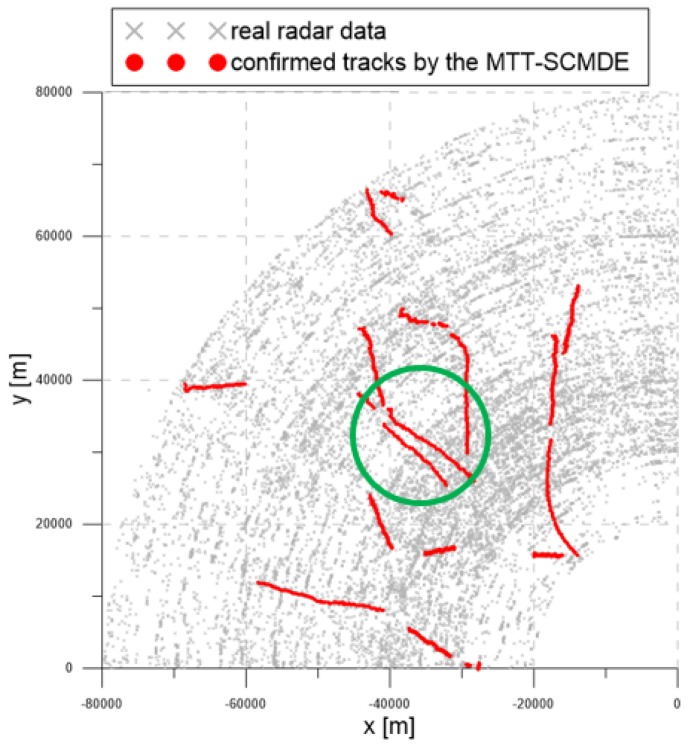
The trajectories of the confirmed tracks by using the proposed MTT-SCMDE.

**Table 1 sensors-20-00114-t001:** Track retention statistics for Monte Carlo simulation.

	True CMD	SCMDE	SCMDE	MTT-SCMDE	MTT-SCMDE
		with *n* = 1	with *n* = 5	with *n* = 1	with *n* = 5
nCase	1199	446	1053	673	1056
nOk (%)	95.91	93.05	95.35	94.11	95.45
nSwitch (%)	1.08	2.91	1.33	2.03	1.42
nMerge (%)	0.59	0.67	0.85	1.10	0.76
nLost (%)	2.42	3.37	2.47	2.76	2.37

**Table 2 sensors-20-00114-t002:** Track retention statistics for Monte Carlo simulation.

	True CMD	SCMDE	SCMDE	MTT-SCMDE	MTT-SCMDE
		with *n* = 1	with *n* = 5	with *n* = 1	with *n* = 5
nCase	1581	485	1324	825	1354
nOk (%)	87.86	74.64	86.25	81.94	87.59
nSwitch (%)	5.76	13.20	6.42	7.52	6.28
nMerge (%)	4.87	8.45	5.66	6.91	4.36
nLost (%)	1.51	3.71	1.67	3.63	1.77

**Table 3 sensors-20-00114-t003:** Track retention statistics for Monte Carlo simulation.

	True CMD	SCMDE	SCMDE	SCMDE	MTT-SCMDE	MTT-SCMDE	MTT-SCMDE
		with *n* = 1	with *n* = 5	with *n* = 7	with *n* = 1	with *n* = 5	with *n* = 7
nCase	2291	398	1876	1881	457	1894	1900
nOk (%)	83.72	64.57	79.26	80.33	67.83	81.73	82.32
nSwitch (%)	8.07	18.59	10.13	10.10	19.26	10.09	9.74
nMerge (%)	7.03	13.82	9.22	7.55	8.97	6.02	5.84
nLost (%)	1.18	3.02	1.39	2.02	3.94	2.16	2.10

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
