# Peer review of "Adaptive Estimation of Spatial Clutter Measurement Density Using Clutter Measurement Probability for Enhanced Multi-Target Tracking"

_sensors, 2019, doi:10.3390/s20010114_

Round 1

Reviewer 1 Report

My main concerns are listed as follows:

(1) In Probabilistic Multi-Hypothesis Tracker (PMHT) algorithm, the number of clutter measurements is assumed to be Poisson distributed with an unknown mean parameter. Then, the parameter can be estimated by the EM approach. Hence, please provide a justification that the spatial clutter measurement density estimator is better or more reasonable in your approach.

(2) In paragraph 2 on page 9, the authors stated: "performance of the existing SCMDE is analyzed for multi-target tracking in homogeneous clutter environments". However, in Appendix A, the clutter is distributed with a Gamma probability density function (pdf) with the number of clutter measurements. Is it reasonable?

(3) In section5, page 10, paragraph 2, line 264-265, the authors stated: "existing SCMDE has two incompatible aspects in tracking performance for multi-target tracking environments". In your adaptive MTT-SCMDE, how to address the two incompatible aspects?

(4) In the SCMDE framework, adjacent measurements are assumed to be generated from clutter. In your adaptive MTT-SCMDE, how to distinguish a measurement originating from the clutter or one of the multiple targets?

(5) In multiple target applications, the authors should provide kinematic state estimation results for a single target.

Reviewer 2 Report

The paper introduces a method for estimation of spatial clutter density in radar measurements. According to the presented results, the proposed method improves the accuracy of multiple targets tracking in non-uniform clutter density scenarios. The Authors have performed extended experiments to prove advantages of the proposed method. The simulation scenarios have taken into account different numbers of targets. The accuracy the proposed method was also verified in experiments with real data obtained from radar.

The list of references in this paper is out of date. Authors have to include the latest publications in this list and update the discussion of related works in Section 1.

Author Response

This manuscript is a resubmission of an earlier submission. The following is a list of the peer review reports and author responses from that submission.

Round 1

Reviewer 1 Report

This paper presents  a solution for  tracking targets in a cultter surveillance

environment, a data association technique  based on clutter measurements. The major weakness of this paper are :

1. Although authors used a simulation scenario of clutter,  the proposed scene is far way from realistic surveillance scenario.   

2. From the  described method,  location information is abstracted from Spatial Clutter Measurement. What is the different  between spatial clutter measurement and  geometric information?   

3. If the objects tracking algorithm based on the geometric centre of the obtained cluster, a variety of classic methods such k-means, means shift method should be investigated as comparison. 

All those above point made me would like to express doubts about the validity of scientific findings  from this paper.

Reviewer 2 Report

The problem of the paper research is very meaningful and difficult, because it is a classic question. The reviewer believes that to solve this problem fundamentally, the only method is to improve the measurement performance of the sensor. Of course, at present, it is still very meaningful to study this problem.

However, this paper has serious flaws in writing, mainly in the following aspects:

1. In the abstract section, the author has written a background on the research, but it is not clear about the proposed method, such as specific implementation techniques.

2. The paper gives an introduction to many methods, but does not clearly explain the key problem. In the process of reading, it has not been found that the method proposed by the author has the possibility of solving the problem.

3. The author uses three Sections,such as Section 2-4 to describe the model and related methods, but only using one section for the proposed method. This has led to a lack of innovation in the paper.

4.In the simulation experiment, which of the five cases is the method proposed in the paper?

Reviewer 3 Report

The Authors have proposed a method, which enables estimation of spatial clutter measurement density for enhanced tracking of multiple targets based on radar or sonar data.

The experimental evaluation of the method presented in the manuscript is insufficient. The experiments have to be extended to prove advantages of the proposed method. The considered simulation scenarios assume very simple two-dimensional straight-line trajectories of three targets that move with constant velocity. The simulation model does not reflect real-world scenarios. Authors have tested the accuracy of their proposed method only via simulation. Additional experiments have to be performed for real data from radar or sonar. Accuracy of the method should be analyzed for various numbers of targets.

Discussion of the results presented in the manuscript has to be extended. For instance, interpretation of the RMSE values in Fig. 9 is not obvious as the results for particular cases are very similar. Thus, additional comments are needed. The same remark relates to the remaining charts.

Round 2

Reviewer 1 Report

The author responded the questions with some definitions of networked radar system. The revised manuscripts is improved with better explanation.  

Reviewer 2 Report

I think the manuscript has been developed. While some writing error still exists, such as in the line 17: ... the causes of for....The author should check it. Moreover, the abstract is too long.